# Study of the Unidimensionality of the Subjective Measurement Scale of Schizophrenia Coping Oral Health Profile and Index: SCOOHPI

**DOI:** 10.3390/bs12110442

**Published:** 2022-11-11

**Authors:** Mohamad Hamad, Nathalie Rude, Mounir Mesbah, Francesca Siu-Paredes, Frederic Denis

**Affiliations:** 1Laboratoire de Recherches Intégratives en Neurosciences et Psychologie Cognitive, Faculté de Santé, Université de Bourgogne Franche-Comté, 25000 Besançon, France; 2Laboratoire de Statistique Théorique et Appliquée (LSTA), Université de Pierre et Marie Curie, 75006 Paris, France; 3Faculté d’Odontologie de Reims, Université Champagne Ardenne, 2 rue du Général Koenig, 51100 Reims, France; 4EA 75-05 Education, Ethique, Santé, Faculté de Médecine, Université François-Rabelais, 37000 Tours, France

**Keywords:** psychometric validation, SCOOHPI, structure validity, unidimensionality, partial credit model, polytomous Rasch model

## Abstract

Background: The Schizophrenia Coping Oral Health Profile and Index (SCOOHPI) scale studies the coping strategies of schizophrenic patients with regard to oral health. The structural validity of this scale is studied has been studied using factor analyses. In this article, we study the unidimensionality of the SCOOHPI scale to use it as an index. Methods: We studied the internal consistency of the items of the SCOOHPI scale. Then, we studied the construct validity. The unidimensionality of the SCOOHPI scale was studied by the partial credit model. Results: The data used in this study come from five hospitals, and the total number of individuals participating in this study is 96, of which 72% are men and 59% are smokers. The SCOOHPI scale has good internal consistency (α = 0.84). The validity of divergence was checked by the absence of correlation between the SCOOHPI scale and the GOHAI (General Oral Health Assessment Index) scale. The unidimensionality of the SCOOHPI scale with data smoothing was demonstrated by the partial credit model. Conclusion: In this study, we completed the study of the psychometric validation of the SCOOHPI. The SCOOHPI scale can then contribute to improving evaluation of the coping strategies of schizophrenic patients with regard to oral health.

## 1. Introduction

Schizophrenia is a mental illness, which affects 1% of the population [1]. Symptoms of schizophrenia are characterized by mental dissociation, hallucinations and inappropriate emotions as well as characterized by episodic or continuous progression [2]. Physical health is affected by mental health and vice versa; therefore, people with schizophrenia (PWS) generally have poor physical health. Among the physical health problems affecting these people, their oral health [3] is particularly affected. Negative symptoms such as apathy and emotional blunting are symptoms responsible for social withdrawal, impaired social performance and/or self-care [4,5]. In this case, for example, daily oral hygiene or regular visits to the dentist take a back seat. This is one of the reasons why there is a higher incidence of caries or periodontal disease in PWS compared to the general population [6].

Coping is the thoughts and behaviors used to deal with stressful situations [7]. Coping is a person’s ability to react to a stressor; coping strategies are then specific to each person [8]. It is also observed that coping strategies also depend on the type of symptoms of the mental illness, the stressful situations and the circumstances experienced by each individual [9]. People with schizophrenia tend to have pervasive cognitive problems (memory and recall), personality alterations, self-esteem problems and social dysfunction. All of these impairments reduce the effectiveness of learning new strategies to improve their general health and especially their oral health [10,11]. In this context, understanding the reactions of an individual suffering from schizophrenia and oral disorders is essential, in order to best support those people with a specific care and prevention process according to their own experiences and representations of oral health but also their anxiety and causes of stress. In other words, oral problems affect the ability of schizophrenic patients to heal [12]. Therefore, it is still important to study the impact of oral health problems on the emotions of people with schizophrenia. Therefore, the development of reliable and valid tools is important to improve the oral health of this population.

The Schizophrenia Coping Oral Health Profile and Index (SCOOHPI) scale was developed in an attempt to assess coping strategies related to oral health deficits in these populations [8]. Coping strategies play an important role in the recovery process of PWS and are used to solve problems in daily life [13]. In addition, coping strategies are associated with both subjective (self-esteem and hopelessness) and objective (symptom severity) domains of recovery [14]. Cronbach’s alpha was 0.85 for the entire SCOOHPI scale. The factor analysis identified three dimensions, Physical Wellness Strategies (α = 0.72, for example item 5; “When I move, I feel good”), Moral Wellness Strategies (α = 0.60, for example item 1, “I am looking for simple pleasures (walk, drink coffee, listen to music, watch TV …)”) and Oral Wellness Access Strategies (α = 0.79, for example item 1, “I feel trapped by my relationship with sugar”) [8]. These three dimensions allow for an assessment of coping processes, assessing the ability to engage in different types of coping responses over time [8], which is consistent with the “perceived ability” model of Kato et al. [15].

Psychometric validation of subjective measurement scales is a necessary step before use. This step concerns all the characteristics of the scale and explores the clinical variables and statistical parameters in order to obtain a robust scale. In the literature, psychometric validation often consists only of calculating Cronbach’s alpha and declaring the scale psychometrically valid if its value is greater than 0.7 (this is the case, among others, of studies [16,17]). Subsequently, some authors added the test–retest to study the temporal stability of responses to questionnaires [18,19]. Finally, more recently, structural validity has been added in most psychometric validation studies.

There are several statistical methods that allow studying the structural validity of a subjective measurement scale. The most commonly used methods are factor analysis and the Rasch model. The objective of factor analysis is to reduce the factors explaining the variables [20]. That is, factor analyses are used to detect the dimensions of the scale or the sets of sub-characteristics studied by the scale. Factor analyses study the multi-dimensionality of the scale.

The Rasch model is used to study the unidimensionality of the scale [21]; in other words, this model shows the importance of calculating the global score of the scale. Two models, belonging to the Rasch model family, are used to study the unidimensionality of polytomous scales: the rating scale model [22] and the partial credit model [23]. The rating scale model is a special case of the partial credit model, which can only be applied when all items have the same number of response modalities, whereas the partial credit model does not require any condition on the number of response modalities for the items.

Psychometric validation is therefore a series of steps where the reliability and validity of the scale are studied in order to obtain an operational scale that is ready to explore the targeted concept.

The unidimensionality of the scale verifies that the overall score of the scale measures the concept intended to be measured [24]. In other words, we study if the concept of the scale can be studied by the overall score of this scale. The absence of unidimensionality verifies that the scale does not study the targeted concept, then the intended concept is deduced from the sub-concepts of the scales. However, the unidimensionality of the scale is not contradictory with the existence of sub-concepts explained by the scale. We used the partial credit model to test the unidimensionality of the SCOOHPI scale, whereas this step was not carried out in the article by Francesca Siu-Paredes et al. [8], where only the sub-concepts of the SCOOHPI scale were studied by factor analyses [25].

In this article, we complete the psychometric validation of the SCOOHPI scale using a sample of 96 schizophrenic patients. In particular, we evaluate the unidimensionality of the scale using the partial credit model [26].

## 2. Methodology

### 2.1. SCOOHPI Scale Items

The SCOOHPI scale contains 18 coping strategy assessment items (See Appendix A), 14 positively formulated items (items 1, 2, 3, 4, 6, 7, 8, 10, 11, 12, 13, 14, 16 and 17) and 4 negatively formulated items (items 5, 9, 15 and 18). The negative items are recoded in reverse to make them positive in the sense of a strategy to improve the daily life of the patients. Each item has five response modalities (never, rarely, sometimes, often and always), coded from 0 to 4, thus making it possible to obtain an overall score between 0 and 72 for the scale as a whole.

### 2.2. Studied Population and Data Collection

Patients were referred to the study investigator through the health professional designated as responsible for the patient’s mental health care. They were informed by the investigator of the nature of the research, its objectives, methodology, duration, expected benefits, constraints and foreseeable risks, in accordance with Article L1122-1 of the Public Health Code. PWS were informed that their data would be computerized, confidential and processed anonymously, and that they could access and rectify it at any time [8].

We made sure to have a diversity of staff based on gender, height, weight, tobacco and place of residence so that our results are quite general.

The study consisted of 72% men, 59% smokers. The average weight is 79 kg, the minimum weight is 50 kg and the maximum weight is 132 kg; the average height is 172 cm, the minimum height is 152 cm and the maximum height is 194 cm. Our sample is composed of 96 schizophrenic patients, from 5 hospitals, 13 patients from the center of Chartreuse, 8 from the center of Châlons, 18 from the center of Tours, 34 from the center of Millau and 23 from the center of Paris.

### 2.3. Dental Investigations

The dental hygiene status was measured by standard dental hygiene, namely Oral Hygiene Index Simplified (OHI-S) [27]. Dental caries was assessed at the dentinal (D3) level, and the DMFT index, based on 32 teeth, was calculated using World Health Organization criteria [28].

### 2.4. Dealing with Missing Data

Before proceeding with psychometric validation, the problem of missing data must be resolved. The problem of missing data hinders statistical analysis. The number of missing values in subjective measurement scales should not normally exceed 5%. To correct for these deficiencies, commonly used methods are simple imputation methods, in which missing values are replaced by the mean or median [29].

In our initial sample, 6 patients did not answer all the items of the SCOOPHI scale. We therefore decided to exclude these patients from the study. For all missing data of the remaining 96 patients, we used the simple mean imputation method.

### 2.5. Psychometric Validation

Psychometric validation contains two main headings, scale reliability and scale validity. The validity of a scale is an essential step in the validation of a scale, it ensures that a scale measures what it is supposed to measure. There are several types of validity. However, we present the psychometric validation by logical steps over time. The very first step is content validity.

#### 2.5.1. Validity of the Scale (the First Two Steps)

*Content validity:* Content validity is the ability of the measurement scale to take into account all the characteristic attributes that it is supposed to assess. At the same time, care must be taken to ensure that these items do not call upon characteristics that one explicitly does not wish to solicit.

According to Lynn (1986), there are two Content Validity Indices, the Content Validity Index for the item (I-CVI: Item-Content Validity Index) and the Content Validity Index for the whole of the scale (S-CVI: Scale-Content Validity Index) [30]. In addition, at least three experts are needed to make this assessment. Davis (1992) proposed a continuum for the evaluation of each item [31] (1: “not relevant”, 2: “somewhat relevant”, 3: “quite relevant”, 4: “highly relevant”). The I-CVI is calculated by the proportion of experts who answered 3 or 4. The S-CVI corresponds to the average of the proportions of the I-CVIs of all the items evaluated by the experts. Lynn states that an I-CVI greater than 0.77 is acceptable [30]. According to several authors, the S-CVI should be greater than or equal to 0.8 [31,32,33].

*Face validity:* Face validity examines the degree of understanding of the questionnaire by the individuals being observed. In other words, it looks at the understanding and acceptance of the subjective measurement scales (or tests) by the individuals. To study Face validity, a question is asked at the end of the questionnaire to assess the scale directly by the individuals under study [34]. We opted for an acceptability questionnaire of the scale. Moreover, the number of missing data is often considered as an indirect index of acceptability. Too much missing data are often characteristic of a low level of understanding of the questionnaire.

#### 2.5.2. Scale Reliability

The reliability of the scale measures the stability of the results. There are three types of reliability: internal consistency, reliability over time and sensitivity to change. These last two aspects will have to be studied later.

*Internal consistency:* Internal consistency indicates how similar the test items are in their content, in other words, whether all the items in the scale are consistent with the concept being studied. At this stage, we ensure that the scale measures what it is supposed to measure.

Internal consistency is measured by calculating Cronbach’s alpha [35], where this coefficient takes a value between 0 and 1. This widely used coefficient has, unfortunately, a controversial interpretation [36].

#### 2.5.3. Validity of the Scale

*Construct validity:* Construct validity makes it possible to better identify the real meaning of the concept measured by the scale by distinguishing it from related concepts. To study construct validity, we use measures that study the same phenomenon from other sources [37].

Convergent validity studies the correlation between the different dimensions of the questionnaire with measures for the same dimensions from other sources.

Divergent validity studies the lack of correlation between the different domains of the questionnaire with measures that cannot have a link with this domain.

*Structure validity:* Structure validity is the stage that examines whether the scale assesses a single concept or whether this concept is composed of sub-concepts. In other words, we study whether our scale is unidimensional or if there are sub-dimensions [38].

In this paper, structure validity is studied using the partial credit model.

*Partial credit model:* The partial credit model (PCM) belongs to the Rasch polytomous model family, and this model is presented by Master [23]. All response modalities are ordered; for example, for an item containing m response modalities, 0 is the least favorable response modality and m − 1 is the most favorable response modality from a psychometric point of view.

The PCM is a unidimensional model that can be used in any study where each item has at least two ordered response modalities and there is an intention to combine the results across items to measure an underlying variable. When an item provides more than two response modalities (e.g., three ordinal modalities: 0, 1 and 2), a score of 1 should not be more likely as ability increases because, beyond some point, a score of 1 becomes less likely than a score of 2. Nevertheless, it follows from the expected order 0 < 1 < 2, … < m_i_, of a set of modalities that the conditional probability of choosing x rather than x − 1 on an item should increase monotonically as the ability increases. The PCM examines the homogeneity of the items, but in the case of multidimensional scales, the PCM does not give indications on the numbers of the dimensions or on the items that are grouped in each of the sub-dimensions.

In the PCM, the item fit indicates whether the items belong to the same dimension or whether there are other sub-dimensions but does not define them. There are two types of fit in the partial credit model, outfit and infit [39], where an item has a good fit if the values of outfit and infit are between 0.5 and 1.5 [40].

### 2.6. Ethical Considerations

This study, named “Quality bis”, was approved by the Committee for the Protection of Persons of the Ile de France region (registration number: 2018-A02043-52). After participants had a complete description of the study, informed consent was obtained from each participant, or from the legal guardians of individuals under guardianship. In the latter case, the patient’s legal guardian(s) signed the informed consent.

The study was registered with www.clinicaltrials.gov (accessed on 1 October 2021) under the number NCT03699501. The SCOOHP questionnaire was developed between June 2016 and November 2018.

## 3. Results

### 3.1. Missing Data

Less than 5% of missing responses are observed for each item, except for item 2 (“I am leaving home”), where 6% of missing responses occur. We then used the mean imputation method to correct the missing data.

### 3.2. Internal Consistency

Cronbach’s alpha was 0.84, above the threshold value suggested by Nunnally [41] of 0.70.

### 3.3. Content Validity

A total of 719 verbatim related to the concept of coping. After excluding non-informative elements because they were redundant or morbid, we retained 277 verbatim. Then, the group of experts proceeded to a final selection in order to obtain an acceptable number of items for transfer to patients. This was followed by a feasibility study with 30 PSS, which enabled us to select 23 items among the 32 [42].

### 3.4. Face Validity

Face validity was performed on 30 individuals in the study. They ensured that the items of the SCOOHPI scale were clear and understandable [42]. Thus, the face validity was confirmed.

### 3.5. Construct Validity

The GOHAI (General Oral Health Assessment Index) scale studies the quality of life related to oral health [43], and the SCOOHPI scale studies the coping strategies related to oral health. Therefore, these two scales have two different concepts.

Consequently, we expect a weak correlation between these two scales.

The mean of the SCOOHPI scale scores is 43.13 and the standard deviation is 13.14, while the mean score of the GOHAI scale is 31.06 and the standard deviation is 7.04. The correlation between the GOHAI scale and the SCOOHPI scale is low (0.09), which confirms that the two scales then measure two different concepts. The divergent validity is confirmed.

### 3.6. Structure Validity

#### 3.6.1. Partial Credit Model

The SCOOHPI scale consists of 18 items with 5 response modalities. At least 90 observations (18 × 5 = 90) are needed to run the partial credit model (number of items multiplied by the number of response modalities) [23]. The number of individuals observed is sufficient to use the partial credit model (96 individuals).

*Item Difficulty:* The easiest item is item 7, while the most difficult item is item 17. Therefore, individuals have the highest probability of giving a favorable response to item 7, and they have the lowest probability of giving a favorable response to item 17 (Table 1, Figure 1).

For each item and each modality, the ability of an individual to respond is calculated, which must be greater than the value of the difficulty of the modality (threshold) being studied to have a sufficient probability of having a response greater than that modality for the item under consideration (Table 1). It can be noticed on several occasions that the difficulty of a response modality i is higher than the difficulty of a response modality i + 1.

*Item characteristic curves:* The characteristic curves of the items show the existence of a curve below the other curves, which shows the existence of certain response modalities rarely selected by the respondents for all items, except for item 6 (Figure 2). In other words, there is always at least one privileged modality out of the five response modalities. This confirms the need for data smoothing.

#### 3.6.2. Partial Credit Model with Data Smoothing

The thresholds for each item must be increasing, following the recommendations of Master [23]. Therefore, we grouped response modalities of the item to obtain increasing thresholds for all the items. This corresponds to a smoothing of the data in order to follow these recommendations of the partial credit model.

For items 1, 2, 3, 4, 10, 11, 12 and 13, we go to four response modalities instead of five; for items 5, 7, 8, 9, 14, 15, 17 and 18, we go to three response modalities instead of five; and for item 16 we go to two response modalities (See Appendix B).

According to Figure 3, we notice that a grouping of response modalities 1 and 2 (rarely and sometimes), for items 1, 2, 3, 4, 10, 12 and 13, leads to graphs where all the response modalities have a probability of being chosen by the individuals. The response modalities 2 and 3 (sometimes and often) were then grouped together for item 11, so that all the response modalities have a probability of being chosen by the individuals. Then, based on the same principle, we grouped response modalities 1, 2 and 3 (rarely, sometimes and often) for items 5, 7, 8, 9, 14, 15, 17 and 18. Similarly, for item 16, we grouped the response modalities 0, 1 and 2 (never, rarely and sometimes) together and the modalities 3 and 4 (often and always) together. On the other hand, all the response modalities were kept for item 6.

*Difficulty of the items:* The easiest item is item 7, while the most difficult item is item 17 (Table 2 and Figure 3). After smoothing the data, it can be seen that all the thresholds are ranked in ascending order. This is in accordance with the recommendations of Master [23].

*Information curves:* According to Figure 4, the information curves of the items have a bell shape, which shows that all the items have good precision. Items 6, 7 and 11 provide more information for low-ability individuals. In other words, items 6, 7 and 11 have a greater influence on the overall SCOOHPI score for individuals with fewer coping strategies. In contrast, items 2, 3 and 17 provide more information for individuals with good ability. That is, items 2, 3 and 17 have a greater impact on the overall SCOOHPI score for individuals with more coping strategies.

The information curve of the scale (Figure 5) shows good overall precision of the scale score and provides the maximum information for individuals with average levels of ability. In other words, the SCOOHPI scale provides a score of coping strategies with good precision. In addition, the responses of individuals with average coping strategies (compared to all individuals) will weigh more heavily on the calculation of the overall score.

*Characteristic curves of the items:* The characteristic curves of the items show that all the response modalities are well represented (Figure 6). In other words, all response modalities have probabilities of being chosen by individuals.

*The fit of the items:* The items of the SCOOHPI scale show a good fit (MSQ outfit and MSQ infit between 0.5 and 1.5) with the partial credit model (Table 3).

### 3.7. Relationship between SCOOHPI and Sociodemographic and Clinical Variables

The SCOOHPI scale is unidimensional after data smoothing, and we studied the overall score according to sociodemographic and clinical variables. The calculation of the overall score was conducted in the same way as usual, i.e., by summing the values of the response modalities of each item for each individual.

The correlations between the SCOOHPI scale and the sociodemographic and clinical variables in Table 4 are almost zero. Since the SCOOHPI score does not follow a normal distribution (according to the Shapiro–Wilk test, *p* = 0.05), we used the Wilcoxon test to compare the means of the scores according to the sociodemographic and clinical variables. According to Table 4, the SCOOHPI score is not affected by any of these variables (according to the Wilcoxon test, *p* > 0.05), nor are the DMFT (Decayed, Missing, Filled teeth) and OHIS (Oral Hygiene Index Simplified) variables. Thus, the SCOOHPI scale is not sensitive to these variables.

Oral health is not always correlated with quality-of-life measurement scales. In other words, we can find a person with poor oral health, and this result had no effect on the quality-of-life perception of this person [44].

## 4. Discussion

### 4.1. Internal Consistency

The Cronbach’s alpha of 0.84 shows that the items of the SCOOHPI scale are similar in their content, so we have a good internal consistency.

### 4.2. Construct Validity

The SCOOHPI scale concerns the concept of coping related to oral health, specific to schizophrenia. No scale in the literature measures the same coping construct specific to schizophrenia patients. Therefore, the convergence validity cannot be studied.

Divergent validity is demonstrated by the lack of correlation between the SCOOHPI scale and the GOHAI scale (correlation = 0.09). These two scales explore two different concepts.

### 4.3. Structure Validity

Structure validity is performed in the article by Francesca Sui-Paredes et al. [8] by factor analysis. This method shows the existence of three sub-dimensions each containing six items: physical well-being strategies, moral well-being strategies and access strategies towards oral well-being. Then, the SCOOHPI scale can be used as a profile producing one score per dimension.

On the other hand, our study of the SCOOHPI scale by the partial credit model shows the unidimensionality of the scale and allows us to obtain an index producing an overall score for all the items. The fit of the SCOOHPI scale items shows that all the items are homogeneous. Therefore, according to the partial credit model with data smoothing, the SCOOHPI scale is a unidimensional scale.

The two statistical models have two different approaches. The partial credit model is a probabilistic model that assumes the unidimensionality of the scale. In contrast, factor analysis studies the sub-concepts of a scale, based on correlations between items. “Item response theory” emerged in the 1950s and 1960s as a reaction to classical test theory. Unlike classical test theory, item response theory is not attached to test scores obtained by random samples, but to individual responses to particular items. These responses are modelled as the result of a stochastic process in which the probability of giving a response of a certain type depends on several parameters that are either person-related or item-related. In this case, Rasch postulates that the items have the same discriminative power but a different level of difficulty [45]. The objectives of these two methods are different but can be complementary. One can seek to obtain only one global score of the scale of the concept studied to facilitate the interpretation of the results (unidimensionality of the scale making it possible to obtain an index). In addition, one can also seek to obtain several scores for the scale studied to bring more finesse and precision to the results (generation of sub-concepts making it possible to obtain a profile). Subjective measurement scales can be designed as both an index and a profile, depending on the interest of researchers and clinicians.

### 4.4. Data Smoothing for the Partial Credit Model

The groupings of response modalities may be justified by the confusion for many patients of the terms rarely, sometimes and often. Moreover, for item 16, which concerns smoking, alcohol and drug addictions, patients’ answers are rather dichotomous. Schizophrenic patients find it difficult to determine their words and would be more apt to give yes/no answers.

Theoretically, the grouping of response modalities is a loss of information, so it is necessary to study carefully how to group response modalities. However, according to our results, there are many response modalities that are rarely chosen, so this procedure was important to ensure that all item response modalities have the probability of being chosen by the individuals. Therefore, grouping the response modalities for the SCOOHPI scale makes individuals’ responses clearer, and we get rid of the rarely chosen modalities.

In the literature, several authors have grouped the response modalities, and in most cases the cause was the existence of response modalities rarely selected by individuals and the inversion of the thresholds, notably Denis et al. [46], Pedersen et al. [47] and Franchignoni et al. [48]. Thus, the grouping of response modalities results in unidimensional scales and thresholds that are well ordered.

### 4.5. Limitations

Even if the number of individuals observed is sufficient (96 schizophrenic patients), it is just at the limit of the number of needed individuals (90 individuals). Firstly, the active queue of patients did not allow obtaining a larger sample; on the other hand, the period of inclusion of the patients being in full confinement related to the pandemic of COVID-19, it was difficult to recruit the patients. In this pandemic context, it is questionable whether the observed data were biased for the SCOOHPI scale. It is obvious that the patients were socially isolated and in lack of both psychiatric and oral care. Thus, it can be assumed that the coping strategies of the patients were particularly solicited. For example, the items “I am leaving home”, “I practice a leisure activity (music, singing, drawing, cinema, walking...)” and “I organize myself to go to the dentist” are affected by the confinement. In particular, the item “I organize myself to go to the dentist” presents the highest difficulty of response according to the partial credit model, which may be linked to the pandemic, which led to a disruption of health care services.

### 4.6. SCOOHPI Scale

Completing the SCOOHPI scale with a population of schizophrenics takes less than 10 min per person in self-administration. To use the SCOOHPI scale with a sample of people with schizophrenia, calculating the overall score for each person consists of adding the values of the modality chosen by the patient for each item.

## 5. Conclusions

Structure validity is studied by several methods such as factor analyses and the partial credit model. Each method has a purpose. The factor analyses determine the dimensions and then detect the sub-concepts studied by the scale. The partial credit model shows whether the overall scale score explains the scale concept. In other words, the partial credit model studies the unidimensionality of the scale.

Face validity and content validity are already performed by Francesca Siu-paredes et al. [8], so in this paper, the study of unidimensionality and construct validity complete the psychometric validation of the SCOOHPI scale.

In the future, it is questionable whether the results of the SCOOHPI scale on another population will be comparable to those we obtained during the COVID-19 pandemic.

Finally, it is hoped that the SCOOHPI scale will contribute to improving evaluation of the oral health of people with schizophrenia by promoting optimal coping strategies, especially to deal with side effects of treatments impacting the perception of oral health and to facilitate therapeutic education of patients for all treatments.

## Figures and Tables

**Figure 1 behavsci-12-00442-f001:**
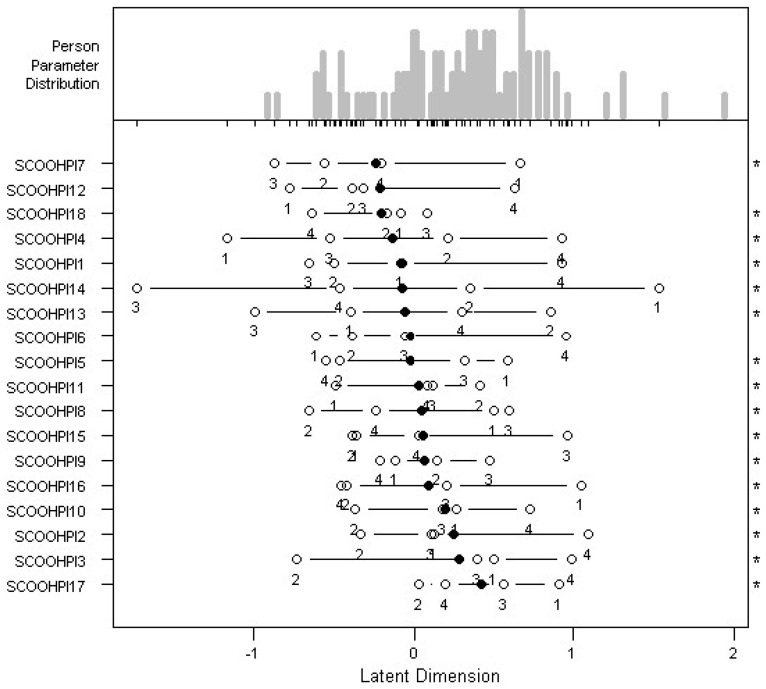
Item difficulty map.

**Figure 2 behavsci-12-00442-f002:**
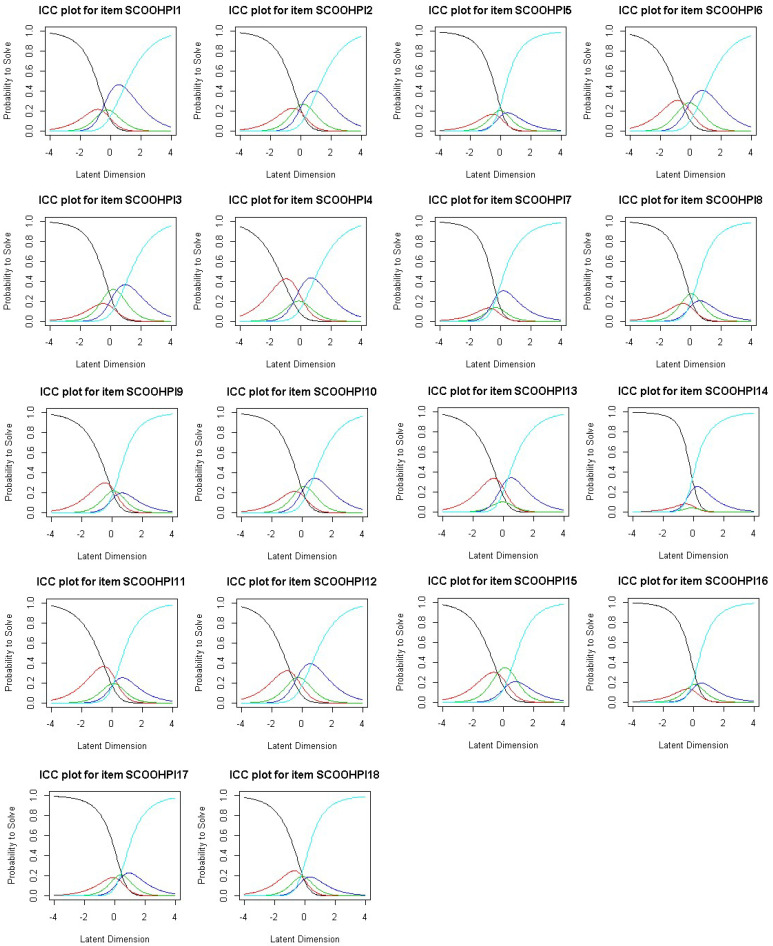
Item characteristic curves.

**Figure 3 behavsci-12-00442-f003:**
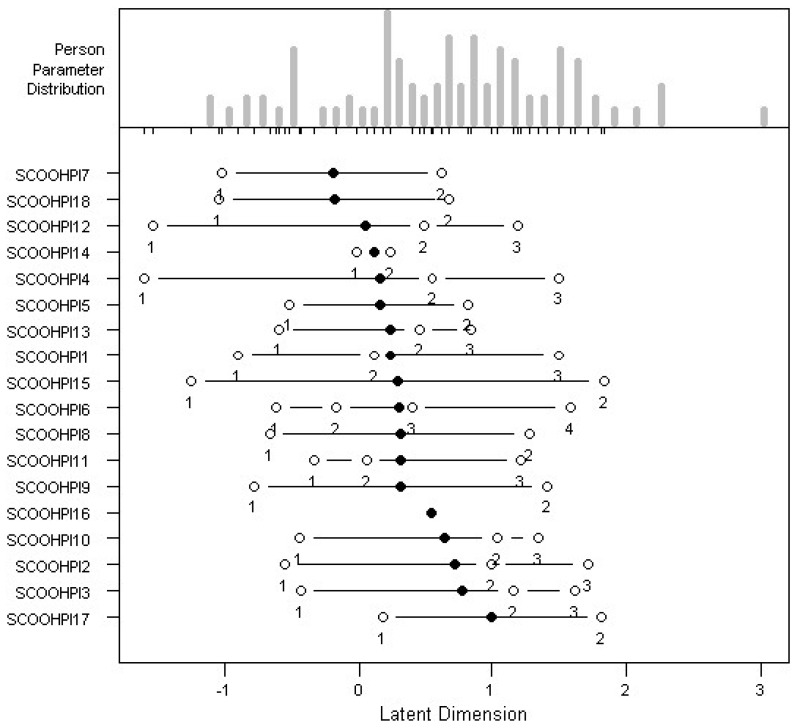
Item difficulty map.

**Figure 4 behavsci-12-00442-f004:**
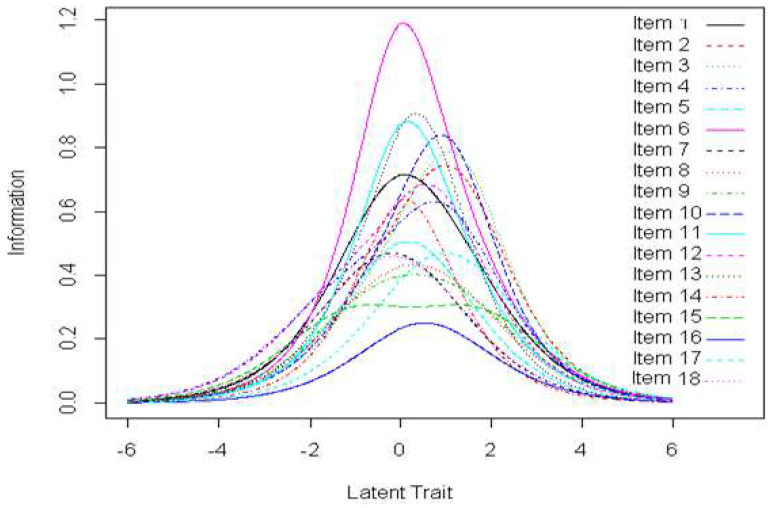
Item information curves.

**Figure 5 behavsci-12-00442-f005:**
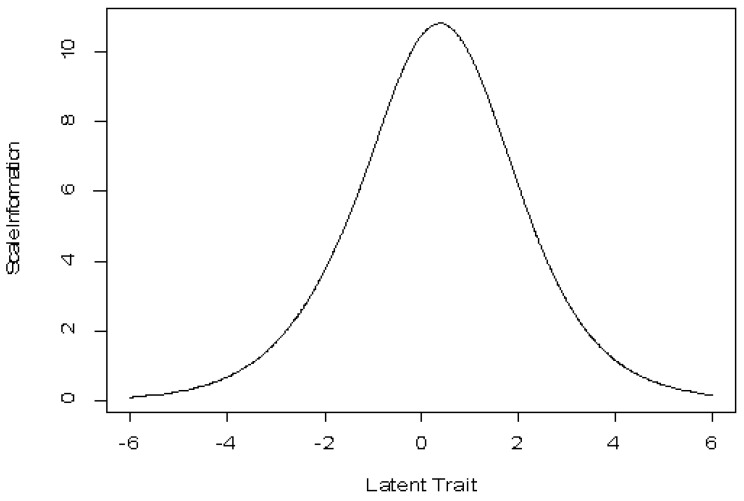
Scale information curve.

**Figure 6 behavsci-12-00442-f006:**
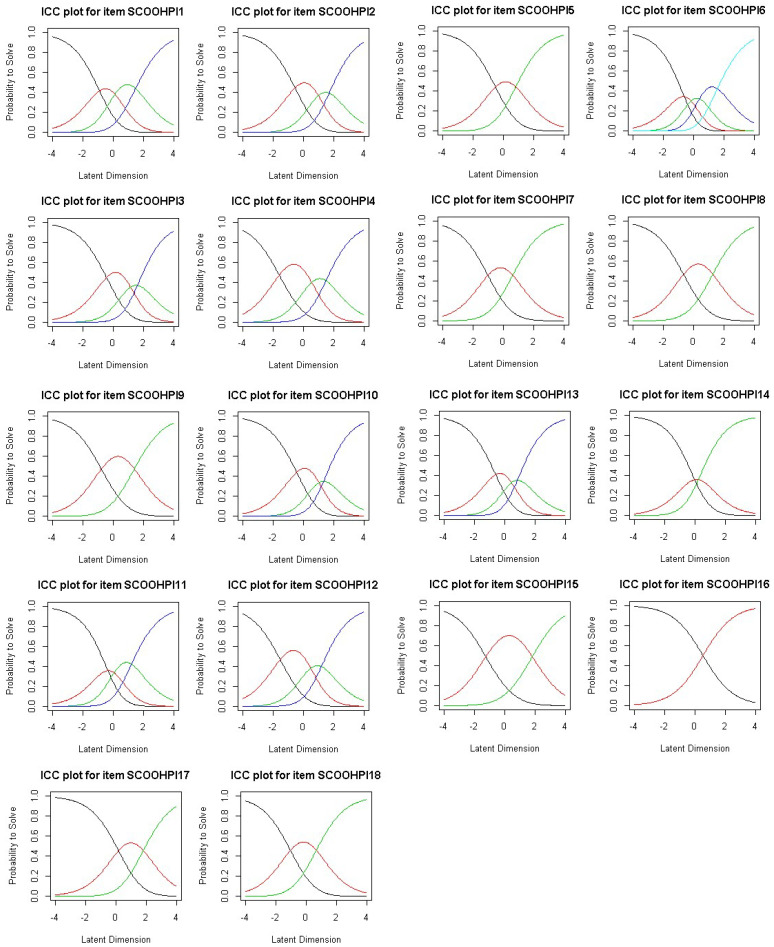
Item characteristic curves.

**Table 1 behavsci-12-00442-t001:** Item difficulty.

	Location	Threshold 1	Threshold 2	Threshold 3	Threshold 4
SCOOHPI1	−0.07708	−0.08235	−0.49956	−0.65266	0.92626
SCOOHPI2	0.25188	0.13451	−0.33656	0.11796	1.09162
SCOOHPI3	0.28787	0.50080	−0.73109	0.39781	0.98394
SCOOHPI4	−0.13658	−1.16237	0.21367	−0.52500	0.92738
SCOOHPI5	−0.02838	0.59144	−0.46858	0.32007	−0.55644
SCOOHPI6	−0.02839	−0.61307	−0.38894	−0.06102	0.94945
SCOOHPI7	−0.24321	0.66568	−0.55963	−0.87389	−0.20501
SCOOHPI8	0.05277	0.50725	−0.65809	0.59843	−0.23652
SCOOHPI9	0.07202	−0.12227	0.14941	0.47958	−0.21863
SCOOHPI10	0.20209	0.26677	−0.37045	0.18009	0.73195
SCOOHPI11	0.03242	−0.49277	0.41420	0.12084	0.08742
SCOOHPI12	−0.21408	−0.78106	−0.38933	−0.31842	0.63248
SCOOHPI13	−0.05909	−0.39943	0.85682	−0.99667	0.30292
SCOOHPI14	−0.07470	1.53286	0.35857	−1.72648	−0.46374
SCOOHPI15	0.06027	−0.35904	−0.38641	0.96121	0.02530
SCOOHPI16	0.09597	1.04624	−0.42061	0.21328	−0.45503
SCOOHPI17	0.42642	0.90643	0.03499	0.56258	0.20169
SCOOHPI18	−0.20367	−0.08559	−0.17496	0.08713	−0.64125

**Table 2 behavsci-12-00442-t002:** Item difficulty.

	Location	Threshold 1	Threshold 2	Threshold 3	Threshold 4
SCOOHPI1	0.23764	−0.89722	0.11623	1.49391	NA
SCOOHPI2	0.71969	−0.55180	0.99642	1.71444	NA
SCOOHPI3	0.78045	−0.43402	1.15878	1.61657	NA
SCOOHPI4	0.15413	−1.59353	0.55462	1.50130	NA
SCOOHPI5	0.15621	−0.51304	0.82545	NA	NA
SCOOHPI6	0.30009	−0.61055	−0.16463	0.39194	1.58362
SCOOHPI7	−0.19463	−1.01576	0.62649	NA	NA
SCOOHPI8	0.30706	−0.66078	1.27490	NA	NA
SCOOHPI9	0.31380	−0.77984	1.40745	NA	NA
SCOOHPI10	0.64687	−0.44213	1.03951	1.34322	NA
SCOOHPI11	0.31273	−0.33621	0.05742	1.21697	NA
SCOOHPI12	0.05061	−1.52920	0.48662	1.19442	NA
SCOOHPI13	0.23520	−0.59128	0.45031	0.84658	NA
SCOOHPI14	0.10930	−0.01401	0.23261	NA	NA
SCOOHPI15	0.29194	−1.24631	1.83019	NA	NA
SCOOHPI16	0.53556	0.53556	NA	NA	NA
SCOOHPI17	0.99623	0.17489	1.81756	NA	NA
SCOOHPI18	−0.17863	−1.03576	0.67851	NA	NA

NA: deleted modalities.

**Table 3 behavsci-12-00442-t003:** Item fit statistics.

	Outfit MSQ	Infit MSQ
SCOOHPI1	1.165	1.097
SCOOHPI2	1.009	1.033
SCOOHPI3	1.267	1.269
SCOOHPI4	0.880	0.895
SCOOHPI5	1.163	1.116
SCOOHPI6	1.060	1.117
SCOOHPI7	0.598	0.606
SCOOHPI8	0.681	0.676
SCOOHPI9	0.921	0.916
SCOOHPI10	0.807	0.827
SCOOHPI11	0.828	0.832
SCOOHPI12	0.876	0.895
SCOOHPI13	0.890	0.866
SCOOHPI14	0.778	0.786
SCOOHPI15	0.904	0.908
SCOOHPI16	1.337	1.229
SCOOHPI17	1.234	1.197
SCOOHPI18	1.064	1.057

**Table 4 behavsci-12-00442-t004:** Relationship between SCOOHPI and sociodemographic and clinical variables.

Variable		SCOOHPI Scale Overall Score	Wilcoxon Test	Correlation with SCOOHPI
Weight	<81 KG	Mean = 26.22 Standard deviation = 7.24	0.51	−0.14
	≥81 KG	Mean = 24.84 Standard deviation = 8.86		
Height	<177 Cm	Mean = 26.39 Standard deviation = 7.90	0.28	−0.05
	≥177 Cm	Mean = 24.28 Standard deviation = 7.77		
Smoking	Yes	Mean = 25.65 Standard deviation = 8.31	0.91	−0.01
	No	Mean = 25.74 Standard deviation = 7.32		
Gender	Male	Mean = 25.33 Standard deviation = 7.71	0.50	0.07
	Female	Mean = 26.59 Standard deviation = 8.38		
DMFT	<15	Mean = 25.28 Standard deviation = 7.81	0.66	−0.03
	≥15	Mean = 26.36 Standard deviation = 8.06		
OHIS	<1.5	Mean = 23.82 Standard deviation = 8.61	0.07	0.12
	≥1.5	Mean = 27.40 Standard deviation = 6.79		

DMFT: Decayed, Missing, Filled teeth; OHIS: Oral Hygiene Index Simplified.

## Data Availability

The data that support the findings of this study are available from Francesca Siu-Paredes upon reasonable request.

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
