# Peer review of "Study of the Unidimensionality of the Subjective Measurement Scale of Schizophrenia Coping Oral Health Profile and Index: SCOOHPI"

_behavsci, 2022, doi:10.3390/bs12110442_

Round 1

Reviewer 1 Report

The manuscript reports a study of the development and validation of the Schizophrenia Coping Oral Health Profile and Index (SCOOHPI). The psychometric properties of the SCOOHPI, including internal consistency, validity and dimensionality, were investigated. The analytical part was well-planned and executed.

However, my main concern for this manuscript is its scientific value and additional contribution to understanding health among individuals with schizophrenia. At this point, the manuscript lacks a clear and focused introduction on the concept it intends to measure and the nature of the concept, which is of extreme importance to scale development and validation. It is not sure of the need to develop such a scale for a specific aspect of health in a focused population of individuals. Indeed, it is clear to clinicians and researchers that individuals with schizophrenia are more vulnerable to poorer physical health and adopted fewer health-promoting behaviours. Then, what is the implication of the existence of SCOOHPI on the well-being of individuals with schizophrenia? A literature review on the relationship between schizophrenia and poorer oral health, and the underlying coping mechanisms would definitely be informative, setting the stage for the need to develop SCOOHPI.

Several additional comments are as follows:

Introduction:

-          As coping is the core of the SCOOHPI. How coping is defined in this setting? My understanding is that “coping” is strategies/ ways to cope with stress/ difficulties (e.g. poorer oral health). Different dimensions of coping are discussed in the literature, e.g. active/ passive, emotional-focused / problem-focused. Are these dimensions relevant to the development of the scale? Is oral health an outcome of coping, or is coping a reaction to poor oral health? The conceptualization of “coping” needs to make explicitly and direct in the Introduction.

-          A previous study identified three dimensions of the SCOOHPI: physical wellness strategies, moral wellness strategies, and oral wellness access strategies. Example items should be provided for each dimension for readers’ consideration. Also, how these dimensions are relevant to the concept it intends to measure? This goes back to my main concern about the manuscript.

-          I would suggest removing the names of the specific scales mentioned in the Introduction when introducing various matrixes of psychometrics (lines 49 – 58, p. 2), as they could distract readers and are not relevant to the SCOOHPI and.

Methods/ Results:

-          There is a lack of a “Participants” section in the manuscript, which is highly critical for studies recruiting clinical samples. I checked the trial registry and the details of the recruitment and inclusion (and exclusion criteria) there should be relevant here.

-          I am curious about the inclusion of the General Oral Health Assessment Index (GOHAI) as a test of construct validity. Section 3.5 mentions that the GOHAI and SCOOHPI “have two different concepts. Consequently, we expect a weak correlation between these two scales” (lines 217-218, p. 5). I am not uncertain if both scales measure distinct and interrelated concepts or distinct but unrelated concepts. This inquiry also hinges on the conceptualization of “coping” (my comment above), and its linkage to oral health. This distinction is critical, which would justify the use of GOHAI as a test for convergent or divergent validity.

-          Similarly, the inclusion of the Decayed, Absent, Filled teeth (DAF) and Oral Hygiene Index Simplified (OHIS) needs more elaboration. It shares the above concern: if the relationships of these scales with the concept that SCOOHPI intends to measure is not clearly stated, it is hard to interpret the result that “the SCOOHPI score is not affected by any of these variables (according to the Wilcoxon test, p>0.05), nor are the DAF (Decayed, Absent, Filled teeth) and OHIS (Oral Hygiene Index Simplified) variables. Thus, the SCOOHPI scale is not sensitive to these variables” (lines 308 – 311, p. 12). It is unsure if the lack of significance could be interpreted as a lack of convergent validity of the scale, which would discredit the psychometric soundness of the scale.

Discussion:

-          The current analysis using the partial credit model supports the unidimensionality of the scale, which is in contrast to the previous study using factor analysis. The authors suggest that “These two different but complementary approaches allow the use of the scale in two different ways, either by obtaining a score per dimension (factor analysis) or by obtaining an overall score (partial credit model). An overall score allows summarizing the differences between two populations, and a score per dimension allows qualifying the conceptual and cultural differences between two populations.” (lines 339 – 243, p. 13). It is no surprise that different statistical approaches give distinct results/ recommendations, however, it boils down to the interpretation: how to make sense of these statistical differences? And more importantly, what are the real-life recommendations for researchers and clinicians to use the scale. These should be stated more directly without ambiguity, which facilitates the use of the scale.

Appendix: The English translation of the items of SCOOHPI should also be provided for international readers.

Author Response

Response to reviewer 1

We would like to thank you for taking the necessary time and effort to review the manuscript. We sincerely appreciate all your valuable comments and suggestions, which helped us in improving the quality of the manuscript.

  1. However, my main concern for this manuscript is its scientific value and additional contribution to understanding health among individuals with schizophrenia. At this point, the manuscript lacks a clear and focused introduction on the concept it intends to measure and the nature of the concept, which is of extreme importance to scale development and validation. It is not sure of the need to develop such a scale for a specific aspect of health in a focused population of individuals. Indeed, it is clear to clinicians and researchers that individuals with schizophrenia are more vulnerable to poorer physical health and adopted fewer health-promoting behaviours. Then, what is the implication of the existence of SCOOHPI on the well-being of individuals with schizophrenia? A literature review on the relationship between schizophrenia and poorer oral health, and the underlying coping mechanisms would definitely be informative, setting the stage for the need to develop SCOOHPI.

Response:

We fully agree with your comment. We have made the requested change. Please see line 41-55 on page 1-2.

  1. As coping is the core of the SCOOHPI. How coping is defined in this setting? My understanding is that “coping” is strategies/ ways to cope with stress/ difficulties (e.g. poorer oral health). Different dimensions of coping are discussed in the literature, e.g. active/ passive, emotional-focused / problem-focused. Are these dimensions relevant to the development of the scale? Is oral health an outcome of coping, or is coping a reaction to poor oral health? The conceptualization of “coping” needs to make explicitly and direct in the Introduction.

Response:

We have made the requested change. Please see line 41-55 on page 1-2.

  1. A previous study identified three dimensions of the SCOOHPI: physical wellness strategies, moral wellness strategies, and oral wellness access strategies. Example items should be provided for each dimension for readers’ consideration. Also, how these dimensions are relevant to the concept it intends to measure? This goes back to my main concern about the manuscript.

Response:

We have made the requested change. Please see line 62-68 on page 2.

  1. I would suggest removing the names of the specific scales mentioned in the Introduction when introducing various matrixes of psychometrics (lines 49 – 58, p. 2), as they could distract readers and are not relevant to the SCOOHPI and.

Response:

Thank you for this suggestion. We have made the changes requested in the manuscript.

  1. There is a lack of a “Participants” section in the manuscript, which is highly critical for studies recruiting clinical samples. I checked the trial registry and the details of the recruitment and inclusion (and exclusion criteria) there should be relevant here.

Response:

We have made the requested change. Please see line 114-119 on page 3.

  1. I am curious about the inclusion of the General Oral Health Assessment Index (GOHAI) as a test of construct validity. Section 3.5 mentions that the GOHAI and SCOOHPI “have two different concepts. Consequently, we expect a weak correlation between these two scales” (lines 217-218, p. 5). I am not uncertain if both scales measure distinct and interrelated concepts or distinct but unrelated concepts. This inquiry also hinges on the conceptualization of “coping” (my comment above), and its linkage to oral health. This distinction is critical, which would justify the use of GOHAI as a test for convergent or divergent validity.

Response:

The GOHAI scale is a functional scale and not a coping scale which does not take into account the coping strategies of individuals. Although valid in schizophrenics, it only partially investigates the disturbances of quality of life related to oral health [Denis, F., Hamad, M., Trojak, B., Tubert-Jeannin, S., Rat, C., Pelletier, J. F., Rude, N. (2017). Psychometric characteristics of the “General Oral Health Assessment Index (GOHAI)» in a French representative sample of patients with schizophrenia. BMC Oral Health, 17(1), 1-10]. The lack of correlation shows that these scales have two different concepts.

  1. Similarly, the inclusion of the Decayed, Absent, Filled teeth (DAF) and Oral Hygiene Index Simplified (OHIS) needs more elaboration. It shares the above concern: if the relationships of these scales with the concept that SCOOHPI intends to measure is not clearly stated, it is hard to interpret the result that “the SCOOHPI score is not affected by any of these variables (according to the Wilcoxon test, p>0.05), nor are the DAF (Decayed, Absent, Filled teeth) and OHIS (Oral Hygiene Index Simplified) variables. Thus, the SCOOHPI scale is not sensitive to these variables” (lines 308 – 311, p. 12). It is unsure if the lack of significance could be interpreted as a lack of convergent validity of the scale, which would discredit the psychometric soundness of the scale.

Response:

We have made the requested change. Please see line 128-132 on page 3 and line 341-344 on page 13.

  1. The current analysis using the partial credit model supports the unidimensionality of the scale, which is in contrast to the previous study using factor analysis. The authors suggest that “These two different but complementary approaches allow the use of the scale in two different ways, either by obtaining a score per dimension (factor analysis) or by obtaining an overall score (partial credit model). An overall score allows summarizing the differences between two populations, and a score per dimension allows qualifying the conceptual and cultural differences between two populations.” (lines 339 – 243, p. 13). It is no surprise that different statistical approaches give distinct results/ recommendations, however, it boils down to the interpretation: how to make sense of these statistical differences?

Response:

We have made the requested change. Please see line 372-388 on page 14.

  1. And more importantly, what are the real-life recommendations for researchers and clinicians to use the scale. These should be stated more directly without ambiguity, which facilitates the use of the scale.

Response:

We have made the requested change. Please see line 420-424 on page 15.

  1. Appendix: The English translation of the items of SCOOHPI should also be provided for international readers.

Response:

Thank you for this remark, the English translation is added in the appendix.

Reviewer 2 Report

The authors developed this study with the aim of completing the psychometric validation study of the SCOOHPI, in order to improve the evaluation of the coping strategies of schizophrenic patients with respect to oral health. To do this, the authors first explained the statistical methodology that allows the psychometric validation of a scale and then presented the data representative of the study of the internal consistency of the scale items and its construct validity.

The quality of the English is good and the organization of the paragraphs is clear and smooth. The introductory paragraphs are detailed, especially as regards the statistical aspect. This is interesting for the reader, who thus has a more detailed picture of the theoretical aspects behind the statistical analyses conducted to achieve the objective of the study.

However, the presentation of the SCOOHPI scale could be further explored, describing its use currently in the literature through major bibliographic citations and presenting or citing some of the research studies that have used it for purposes similar to those of the authors.

The data (well-presented and represented through the figures) are adequately exposed. In fact, the results section is well structured and divided into sub-sections consistent with the structure of the paper. The clarity of the exposition and organization of the text promotes the reader's understanding.

However, the "Discussion" paragraph should be deepened. In this section it seems that the authors limit themselves to exposing and discussing their data, without many references to other literature. Perhaps, it would be more appropriate to compare one's data in greater depth with those present in the literature and present more clearly how and if the achievement of their objective influences the state of research currently known, citing studies that have used the same tool and comparing the results obtained with the results present in other researches.

Author Response

We would like to thank you for taking the necessary time and effort to review the manuscript. We sincerely appreciate all your valuable comments and suggestions, which helped us in improving the quality of the manuscript.

  1. However, the presentation of the SCOOHPI scale could be further explored, describing its use currently in the literature through major bibliographic citations and presenting or citing some of the research studies that have used it for purposes similar to those of the authors.

Response:

We fully agree with your comment. We have made the requested change. Please see line 41-55 on page 1-2.

  1. However, the "Discussion" paragraph should be deepened. In this section it seems that the authors limit themselves to exposing and discussing their data, without many references to other literature. Perhaps, it would be more appropriate to compare one's data in greater depth with those present in the literature and present more clearly how and if the achievement of their objective influences the state of research currently known, citing studies that have used the same tool and comparing the results obtained with the results present in other researches.

Response:

The SCOOHPI scale is developed in 2018, so this scale is not yet used on another population.

Round 2

Reviewer 1 Report

Thank you for the authors’ effort to address my comments and revise the manuscript. The manuscript is now much improved, especially the Introduction and Discussion. I have two minor recommendations, as follows:

1.      The Introduction could be further beefed up with a few statements that in what ways the oral health of individuals with schizophrenia is compromised and the underlying reasons (e.g. negative symptoms make them less motivated to take care of their own oral health), probably at the end of the first paragraph.

2.      In lines 346-347 (p. 13), the meaning of the statement “In contrast, oral health inequalities are strongly correlated with social inequalities.” Is not clear. Please delete if it is not necessary here, or rephrase.

Author Response

Response to reviewers

Thank you for the authors’ effort to address my comments and revise the manuscript. The manuscript is now much improved, especially the Introduction and Discussion.

Response

Thank you

I have two minor recommendations, as follows:

  1. The Introduction could be further beefed up with a few statements that in what ways the oral health of individuals with schizophrenia is compromised and the underlying reasons (e.g. negative symptoms make them less motivated to take care of their own oral health), probably at the end of the first paragraph.

Response:

      Thank you for this suggestion. We have added a new paragraph page 1 lines 40 to 44. 

      We have added 3 more references which we have listed in the "reference" section

      4-Persson K, Olin E, Ostman M. Oral health problems and supportas experienced by people with severe mental illness living in community-based subsidised housing a qualitative study. Health Soc Care Community. 2010; 18:529- 36.

      5-Kilbourne AM, Horvitz-Lennon M, Post EP, McCarthy JF, Cruz M,Welsh D, et al. Oral health in Veterans Affairs patients diagnosed withserious mental illness. J Public Health Dent. 2007; 67:42- 8.

      6- Denis F, Goueslard K, Siu-Paredes F, Amador G, Rusch E, Bertaud V, Quantin C. Oral health treatment habits of people with schizophrenia in France: A retrospective cohort study. PLoS One. 2020 Mar 9;15(3):e0229946. doi: 10.1371/journal.pone.0229946. PMID: 32150582; PMCID: PMC7062238.

  1. In lines 346-347 (p. 13), the meaning of the statement “In contrast, oral health inequalities are strongly correlated with social inequalities.” Is not clear. Please delete if it is not necessary here, or rephrase.

Response

We agree that this sentence is confusing. We have deleted it as you suggested.

We would like to sincerely thank the reviewers for taking the time to review our manuscript and for all their very constructive comments